# Learning the non-Markovian features of subsystem dynamics

**Michele Coppola**[1][⋆], **Mari Carmen Bañuls**[2,3][•] **and Zala Lenarčič**[1][†]

**1** Jožef Stefan Institute, 1000 Ljubljana, Slovenia
**2** Max-Planck-Institut für Quantenoptik, Hans-Kopfermann-Str. 1, D-85748 Garching,
Germany
**3** Munich Center for Quantum Science and Technology (MCQST), Schellingstr. 4, D-80799
München, Germany

⋆ michele.coppola@ijs.si , • banulsm@mpq.mpg.de , † zala.lenarcic@ijs.si

## Abstract

The dynamics of local observables in a quantum many-body system can be formally described in the language of open systems. The problem is that the bath representing the complement of the local subsystem generally does not allow the common simplifications often crucial for such a framework. Leveraging tensor network calculations and optimization tools from machine learning, we extract and characterize the dynamical maps for single- and two-site subsystems embedded in an infinite quantum Ising chain after a global quench. We consider three paradigmatic regimes: integrable critical, integrable non-critical, and chaotic. For each we find the optimal time-local representation of the subsystem dynamics at different times. We explore the properties of the learned time-dependent Liouvillians and whether they can be used to forecast the long-time dynamics of local observables beyond the times accessible through direct quantum many-body numerical simulation. Our procedure naturally suggests a novel measure of non-Markovianity based on the distance between the quasi-exact dynamical map and the closest CP-divisible form and reveals that criticality leads to the closest Markovian representation at large times.

# 1   Introduction

When explaining the long-time behavior of out-of-equilibrium isolated many-body systems, a common interpretation of equilibration of local observables is to view the measured subsystem as an open quantum system, coupled to baths through its boundary [1–4]. Although the entire system remains isolated and in a pure state during the whole evolution (assuming it started in one), its local properties can be captured by an ensemble or mixed state description, obtained by partitioning the system into the measured subsystem and its complement. For generic models, the latter should act as a bath, capable of inducing thermalization of observables to the thermal values corresponding to the initial energy density. While pure state dynamics of the whole system is expected to show monotonic growth of entanglement entropy, the description of the subsystem as a mixed state is expected to generically show a rise and fall of complexity (as measured by the operator entanglement entropy or other proxies) in time [5, 6], while still predicting the dynamics of local observables. Focusing on the relevant degrees of freedom of the subsystem has thus become a compelling line of research in quantum many-body systems [7–15].

If the subsystem is initially unentangled with its complement, the subsystem dynamics is formally described by a completely positive dynamical map [16], obtained by tracing out the degrees of freedom of the complement. While accessing the reduced dynamical map is feasible for quadratic quantum systems linearly coupled to a Gaussian environment [17], this remains a challenging task for generic quantum many-body systems, despite promising theoretical advances based on thermo-field dynamics and the Schwinger-Keldysh formalism [18]. Even from a numerical standpoint, the difficulty lies in simulating many-body interacting dynamics for long times and handling a large number of Kraus operators [16]. Nonetheless, this problem has been numerically addressed for a small number of interacting fermionic modes coupled to non-interacting fermionic baths [19]. In addition, transverse folding tensor network methods [7, 10, 11, 20–22] offer a natural way to approach this task for one-dimensional quantum many-body systems and short-range Hamiltonians, by representing the evolved subsystem at a certain time as a two-dimensional tensor network [Fig. 1] where the effect of environment is encoded in the left and right boundary vectors. The entanglement of these vectors (temporal entanglement) can in certain scenarios grow much slower than the entanglement in the

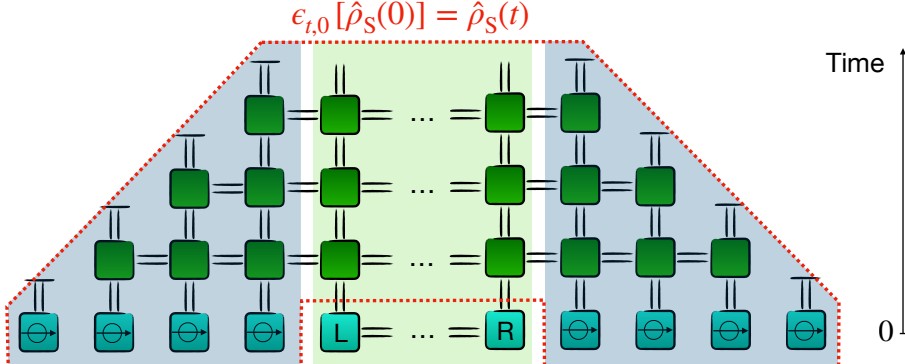

Figure 1: Pictorial representation of the transverse folding tensor network which evolves the subsystem density matrix (spanning between L and R site) up to a given time. Environment is initialized in a product state, with a local state denoted with symbol ⊖ .

physical state, facilitating the numerical treatment [20, 23–27]. However, such a complete description of the environment is generically also limited to only transient times. Therefore it is important to understand whether some further simplification and physical interpretation of the left and right environments is possible. In particular, we are interested in whether—and under what conditions—it is possible to obtain an effective description of the dynamics involving only the local degrees of freedom of the subsystem.

In this work, we focus on a simple, yet phenomenologically rich, one dimensional quantum model, which allows us to analyze the properties of the subsystem dynamics for very different regimes of the full system. We consider a tripartition of an infinite quantum Ising chain, that defines a subsystem consisting of $l$ contiguous sites and two semi-infinite Ising chains, which are treated as an external environment. We examine the subsystem's dynamical maps—obtained via the transverse folding tensor network method and found to be non-divisible—across three case studies: a critical, an integrable non-critical, and a chaotic quantum Ising chain. We then address the following questions: (i) What is the best time-local approximation of the evolution? (ii) Do we need non-Markovian features of the bath to correctly represent the subsystem dynamics? (iii) How do non-Markovian features depend on the dynamical regime (being chaotic, integrable or critical)? This research is motivated by three main motivations. Firstly, if the subsystem dynamics can be (at least approximately) described by a time-local form that can be extrapolated to times going beyond the reach of the whole system (tensor network) evolution, one could utilize computational approaches for open system dynamics, acting on a smaller subsystem Hilbert space and not suffering from the limitations due to the entanglement growth. Second, pseudo-Lindblad forms (8) provide a clearer physical interpretation of dissipation, characterized by Lindblad operators and their corresponding jump rates. Finally, identifying the dissipation channels offers valuable insights for a first-principles derivation of open dynamics, which has not been addressed much in the strong subsystem-environment coupling regimes.

The paper is organized as follows. In Sec. 2, we define the model and its subsystem non-unitary dynamics. In Sec. 3, we present the transverse folding approach [Fig. 1] to obtain the subsystem's dynamical maps $\epsilon_{t,0}$. In Sec. 4, we investigate the divisibility of $\epsilon_{t,0}$, which is crucial for the existence of time-local generators. As the dynamical map $\epsilon_{t,0}$ is singular, i.e., not divisible, we define the optimization problem and employ machine learning techniques to determine the time-local forms that best capture the subsystem dynamics, similar to what has been previously done on learning from the local observables in Refs. [28–30]. In Sec. 5, we present the results of such optimization, demonstrating that our approach effectively interpo-

lates the evolution of local observables. Moreover, we explicitly showcase the dissipation patterns, governed by time-dependent Lindblad operators and use that to forecast the dynamics at longer times. Sec. 6 addresses the characterization of memory effects [31–33] by introducing a new measure of non-Markovianity based on the distance between the quasi-exact evolution and the closest CP-divisible form. In Sec. 7, we present our conclusions and suggest possible directions for future research. Appendices contain technical details and background.

## 2   Model and subsystem dynamical map

The Hamiltonian of an infinite quantum Ising chain reads

$$\hat{H} = \sum_i \left( J \hat{\sigma}_i^z \hat{\sigma}_{i+1}^z + g^x \hat{\sigma}_i^x + g^z \hat{\sigma}_i^z \right), \tag{1}$$

where $\hat{\sigma}$'s are the Pauli matrices, $J$ is the coupling constant, $g^x$ and $g^z$ play the role of transverse and longitudinal fields, respectively. As illustrated in Fig. 1, we consider a partition of the chain in three components, yielding the $l$-site Ising chain as the subsystem (S) coupled at left (L) and right (R) boundaries to two semi-infinite Ising chains, which will be treated as an external environment (E). Therefore, the total subsystem-environment Hamiltonian (1) can be written as $\hat{H} = \hat{H}_S + \hat{H}_E + \hat{H}_I$, with

$$\hat{H}_S = \sum_{i=L}^{R-1} J \hat{\sigma}_i^z \hat{\sigma}_{i+1}^z + \sum_{i=L}^{R} \left( g^x \hat{\sigma}_i^x + g^z \hat{\sigma}_i^z \right), \qquad \hat{H}_I = J \left( \hat{\sigma}_{L-1}^z \hat{\sigma}_L^z + \hat{\sigma}_R^z \hat{\sigma}_{R+1}^z \right), \tag{2}$$

and $\hat{H}_E = \hat{H} - \hat{H}_S - \hat{H}_I$. The Hamiltonian operators $\hat{H}_S$ and $\hat{H}_E$ respectively determine the internal subsystem and environment dynamics, and $\hat{H}_I$ is the interaction Hamiltonian. Let $\hat{\rho}$ be the density operator for the total system. Therefore, $\hat{\rho}_S = \text{tr}_E(\hat{\rho})$ represents the reduced subsystem state, obtained by tracing over the environment degrees of freedom, while $\hat{\rho}_E = \text{tr}_S(\hat{\rho})$ is the environment density matrix. The reduced density matrix $\hat{\rho}_S$ fully characterizes the subsystem, as it determines the expectation value of any local observable $\hat{O}_S \otimes \hat{\mathbb{I}}_E$. The main goal of this work is to characterize the subsystem dynamics and the non-Markovian effects resulting from the strong coupling with the environment ($J \sim g^x, g^z$).

From a formal theoretical perspective, the time-evolved reduced density operator is obtained by solving the Liouville equation for the total density operator $\hat{\rho}$ and tracing out the environmental degrees of freedom

$$\hat{\rho}_S(t) = \text{tr}_E \left( e^{-i\hat{H}t} \hat{\rho}(0) e^{i\hat{H}t} \right). \tag{3}$$

As is common in the theory of open quantum systems [16], we assume that the subsystem and environment are initially uncorrelated, i.e., $\hat{\rho}(0) = \hat{\rho}_S(0) \otimes \hat{\rho}_E(0)$, which implies that the reduced density operator evolves as

$$\hat{\rho}_S(t) = \epsilon_{t,0}[\hat{\rho}_S(0)], \tag{4}$$

for a completely positive superoperator $\epsilon_{t,0}$ acting on the reduced space of the subsystem degrees of freedom [Fig. 1]. This superoperator, known as the *reduced dynamical map*, determines the time evolution of the subsystem state.

Under fairly general conditions, the evolution of $\epsilon_{t,0}$ is generated by the memory kernel $\mathcal{K}_{t,\mu}$ [34, 35], such that

$$\dot{\epsilon}_{t,0} = \int_0^t d\mu \, \mathcal{K}_{t,\mu} \epsilon_{\mu,0}. \tag{5}$$

If the dynamical map is invertible, memory-kernel master equations can be rewritten in the time-local form

$$\dot{\epsilon}_{t,0} = \Lambda_t \epsilon_{t,0}, \tag{6}$$

for the time-local representation

$$\Lambda_t = \int_0^t d\mu \, \mathcal{K}_{t,\mu} \epsilon_{\mu,0} \epsilon_{t,0}^{-1} = \dot{\epsilon}_{t,0} \epsilon_{t,0}^{-1}, \tag{7}$$

which can always be put in the pseudo-Lindblad form [36]

$$\Lambda_t[\hat{\rho}_S(t)] = -i \sum_{i=1}^{d^2-1} h_i(t) [\hat{G}_i, \hat{\rho}_S(t)] + \sum_{ij=1}^{d^2-1} c_{ij}(t) \left( \hat{G}_i \hat{\rho}_S(t) \hat{G}_j - \{\hat{G}_j \hat{G}_i, \hat{\rho}_S(t)\} \right), \tag{8}$$

where $d = 2^l$ is the dimension of the subsystem Hilbert space. The elements $\{\hat{G}_i\}_{i=0,\dots,d^2-1}$ denote an Hermitian orthonormal basis of the Liouville subsystem space, namely $\hat{G}_i = \hat{G}_i^\dagger$ and $\mathrm{tr}_S\{\hat{G}_i \hat{G}_j\} = \delta_{ij}$, with $\hat{G}_0 = d^{-1/2} \hat{\mathbb{1}}_S$. In this work, we choose as a local basis on each site $k \in [L, R]$ the corresponding normalized Pauli matrices, $2^{-1/2}(\hat{\mathbb{1}}_k, \hat{\sigma}_k^x, \hat{\sigma}_k^y, \hat{\sigma}_k^z)$, so that the basis elements $\{\hat{G}_i\}_{i=1,\dots,d^2-1}$ consist of properly normalized Pauli strings. The terms $h_i(t)$ are time-dependent real-valued functions and

$$\hat{H}_{\mathrm{eff}}(t) = \sum_{i=1}^{d^2-1} h_i(t) \hat{G}_i, \tag{9}$$

plays the role of an effective time-dependent Hamiltonian. The coefficient matrix $c_{ij}(t)$ is a $(d^2-1) \times (d^2-1)$ Hermitian matrix and its diagonalization yields $(d^2-1)$ time-dependent real eigenvalues $\gamma_i(t)$ and Lindblad operators $\hat{L}_i(t)$. It is worth noting that, in the standard Lindblad formalism [16], the matrix $c_{ij}$ is time-independent and positive semi-definite, ensuring non-negative eigenvalues $\gamma_i \geq 0$, which are typically referred to as *jump rates* in the context of quantum jump techniques [37]. However, in the generalized time-dependent structure (8), the eigenvalues $\gamma_i(t)$ may take negative values, while still preserving the complete positivity of the full dynamical map $\epsilon_{t,0}$ at all times.

If $\epsilon_{t,0}$ is not invertible, the dynamics cannot typically be expressed in a time-local form, i.e. the map is non-divisible [38]—meaning it is not possible to find a propagator $\epsilon_{t+dt,t} = (1 + \Lambda_t dt)$ such that $\epsilon_{t+dt,0} = \epsilon_{t+dt,t} \epsilon_{t,0}$, for an infinitesimal time step $dt$ and any time $t$. [1] In such cases, the kernel structure in Eq. (5) cannot be bypassed. In a discretized time setting, a possible way to describe the (non-Markovian) dynamics of the subsystem is provided by transfer tensors [40], which encode the indivisible part of the evolution.

## 3  Numerical approximation of the dynamical maps

As previously explained, in this work we consider a quantum Ising chain in the thermodynamic limit, and identify the subsystem of interest with $l$ consecutive sites in the center, and the environment with the two semi-infinite subchains to the sides. We furthermore fix the initial state of the environment to be a translationally invariant product state. In order to explicitly extract the dynamical maps $\epsilon_{t,0}$ at different times [Fig. 1], one would have to compute how the evolution maps each possible initial configuration of the subsystem to a final one, after simulating

---

[1]More rigorously, time-local representations still exist in exceptional cases where the kernel of singular dynamical maps grows monotonically over time; however, this is not the case for the dynamical maps studied in this work [39].

the full system dynamics and tracing out the environment. A convenient tensor network (TN) alternative to the direct simulation of the full evolution is provided by the transverse folding strategy [7, 20, 21]. In this approach, we build a two-dimensional TN representation of the time-evolved state [Fig. 1] by applying a certain number of layers on the initial state, each one representing one step of trotterized time-evolution. The time-dependent expectation value of an observable can be represented by the contraction of the corresponding operator between such TN and its adjoint, so that the problem of simulating the evolved state is traded, instead, by the problem of finding an efficient approximation to the contraction of the resulting network. The contraction of each semi-infinite half-network on the sides of the subsystem (indicated by the blue shadowed areas in Fig. 1) can be interpreted as a vector along the space (horizontal) direction, which gives rise to the concept of temporal entanglement [21]. In the transverse folding strategy, this vector is approximated by a matrix product state (MPS). For a local Hamiltonian, the half-network has a light cone structure, which can be exploited to find a more efficient contraction strategy [41, 42]. After the so-called boundary vectors (also called influence matrices [22]) are found, the expectation values of local observables can be computed contracting the transfer operators for the subsystem sites between the left and right boundaries. If no operator is inserted, and the indices corresponding to the subsystem are left open (see the green shadowed region in Fig. 1), the result is the reduced density matrix for the subsystem after the evolution time.

Because the environment approximation is independent of the subsystem's evolution, the transverse folding algorithm can be immediately adapted to extract the dynamical map $\epsilon_{t,0}$ for a small subsystem at different times. In particular, throughout this work, we prepare the environment in the product state $\rho_{\mathrm{E}}(0) = \otimes_{i<L, i>R} \left| X_i^+ \right\rangle \left\langle X_i^+ \right|$, where $\sigma_i^x \left| X_i^+ \right\rangle = \left| X_i^+ \right\rangle$ (in Fig. 1 $\left| X_i^+ \right\rangle$ is represented by the symbol $\ominus$). We then extract the reduced dynamical maps for subsystems of size $l = 1$ and $l = 2$, using a bond dimension $D = 1000$. This allows us to explore relatively long times with moderate computational resources. The TN can be contracted using the same strategy, with a computational overhead factor proportional to the dimension of the subsystem density operator. In principle, this overhead can be reduced at the expense of further approximating the contraction of the resulting network with a MPO ansatz. However, for the small sizes considered here, it is possible to deal with the full dimension of the reduced density operator, avoiding the additional truncation.

## 4 Learning protocol

Hereinafter, we consider the Hamiltonian (1), with fixed $J = 1.0$, and present the study of three paradigmatic cases: a critical ($g^x = 1.0$, $g^z = 0.0$), integrable non-critical ($g^x = 1.5$, $g^z = 0.0$) and chaotic ($g^x = -1.05$, $g^z = 0.5$) quantum Ising chain. Our numerical simulations showcase that the subsystem's dynamical map $\epsilon_{t,0}$ becomes singular during time evolution and formal time-local representations do not exist. These singularities occur when at least one of the eigenvalues $e_i(t)$ in the spectrum of the subsystem's dynamical map vanishes. The time evolution of the individual eigenvalues for the single-site dynamical maps reveals qualitative differences between the different dynamical regimes. This behavior is illustrated in Fig. 2, which shows the time evolution of the modulus of the complex eigenvalues of single-site dynamical maps across the three representative cases: the critical, integrable non-critical, and chaotic regimes. The modulus of these eigenvalues is plotted on a log-log scale, with a semi-log scale shown in the insets to highlight the different decay behaviors—polynomial or exponential—exhibited by each eigenvalue. The eigenvalues are ordered such that $|e_i(t)| > |e_{i+1}(t)|$, with $i = 1, 2, 3$. In Fig. 2, $|e_1(t)|$ is omitted since, by construction, $|e_1(t)| = 1 \ \forall t$. We observe that in the critical regime [Fig. 2(a)], the second largest eigenvalue decays smoothly as a power

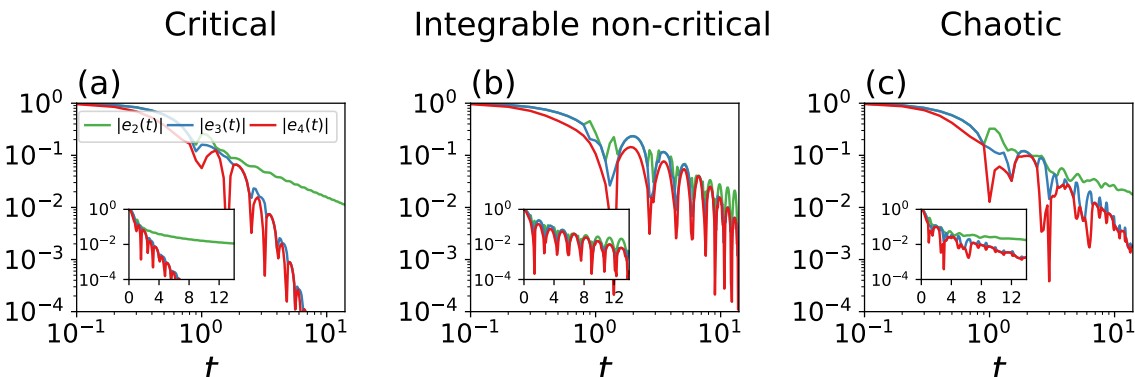

Figure 2: Modulus of the eigenvalues $e_i(t)$ of $\epsilon_{t,0}$ as a function of time in a log-log scale. Here, we present the results for single-site subsystems ($l = 1$) and the (a) critical ($g^x = 1.0$, $g^z = 0.0$), (b) integrable non-critical ($g^x = 1.5$, $g^z = 0.0$) and (c) chaotic ($g^x = -1.05$, $g^z = 0.5$) parameters. In the insets, the same quantities in a semi-log scale. Since $|e_1(t)| = 1$ by construction, $|e_1(t)|$ is omitted.

law, and the singular behaviour of the map arises from $e_3(t)$ and $e_4(t)$, vanishing exponentially fast. In contrast, for the integrable non-critical case [Fig. 2(b)], the decay of $e_3(t)$ and $e_4(t)$, while compatible with an exponential envelope, is much slower, and $e_2(t)$ shows oscillations through zero, contributing to the singularities of the map. For the chaotic case [Fig. 2(c)] we observe a behaviour resembling the critical case, in which the second largest eigenvalue also decays as a power law without large oscillations and $e_3(t), e_4(t)$ decrease exponentially, but at a slower rate.

Despite the non-invertibility of the maps, one can seek the best time-local approximations to the dynamics and evaluate their accuracy through the evolution of quantum observables. To find the best time-local approximation, we define the loss function

$$\text{loss}(t, \Lambda_t) \equiv \|\dot{\epsilon}_{t,0} - \Lambda_t\,\epsilon_{t,0}\|, \tag{10}$$

where $\|\mathcal{O}\| = \sqrt{\text{tr}(\mathcal{O}^\dagger\mathcal{O})}$ is the Frobenius norm of the matrix $\mathcal{O}$ and Eq. (10) represents the Frobenius distance between $\dot{\epsilon}_{t,0}$ and $\Lambda_t\,\epsilon_{t,0}$. [2] Therefore, the optimal time-local form $\Lambda_t^{\text{OPT}}$ is obtained by minimizing the loss function (10) over the space of $(d^2-1)\times(d^2-1)$ Hermitian coefficient matrices $\boldsymbol{c}(t) = \boldsymbol{c}(t)^\dagger$ with elements $c_{ij}(t)$ and real vectors $\boldsymbol{h}(t) = (h_1(t), \ldots, h_{d^2-1}(t))$, Eq. (8), at any time $t$. For a more structured analysis, we define the error function,

$$\mathcal{E}(t) \equiv \text{loss}(t, \Lambda_t^{\text{OPT}}), \tag{11}$$

which is the minimum of the loss function (10) at time $t$ for a large but fixed number of optimization steps. Assuming that the numerical optimization finds the true minimum, the error function (11) should vanish when the dynamical map is divisible, as Eq. (6) implies $\Lambda_t^{\text{OPT}} = \dot{\epsilon}_{t,0}\epsilon_{t,0}^{-1}$.

To minimize the loss function (10), we use the ADAM optimization algorithm [43], an adaptive gradient-based method widely employed in machine learning. Throughout this work, we set the default learning rate $l_r = 10^{-2}$, adjusting it as needed for fine-tuned analysis. Unless stated otherwise, we used $3 \times 10^4$ ADAM optimization steps for single-site subsystems ($l = 1$) and $1.6 \times 10^3$ ADAM optimization steps for two-site subsystems ($l = 2$). For a study of the behavior of the error $\mathcal{E}(t)$ as a function of the ADAM optimization steps, the reader is referred to the Appendix A. After minimizing the loss function (10), the optimized dynamical map $\epsilon_{t,0}^{\text{OPT}}$

---
[2] The derivative $\dot{\epsilon}_{t,0}$ is numerically evaluated as $\dot{\epsilon}_{t,0} \approx (\epsilon_{t+dt,0} - \epsilon_{t,0})/dt$, with time step $dt = 0.1$.

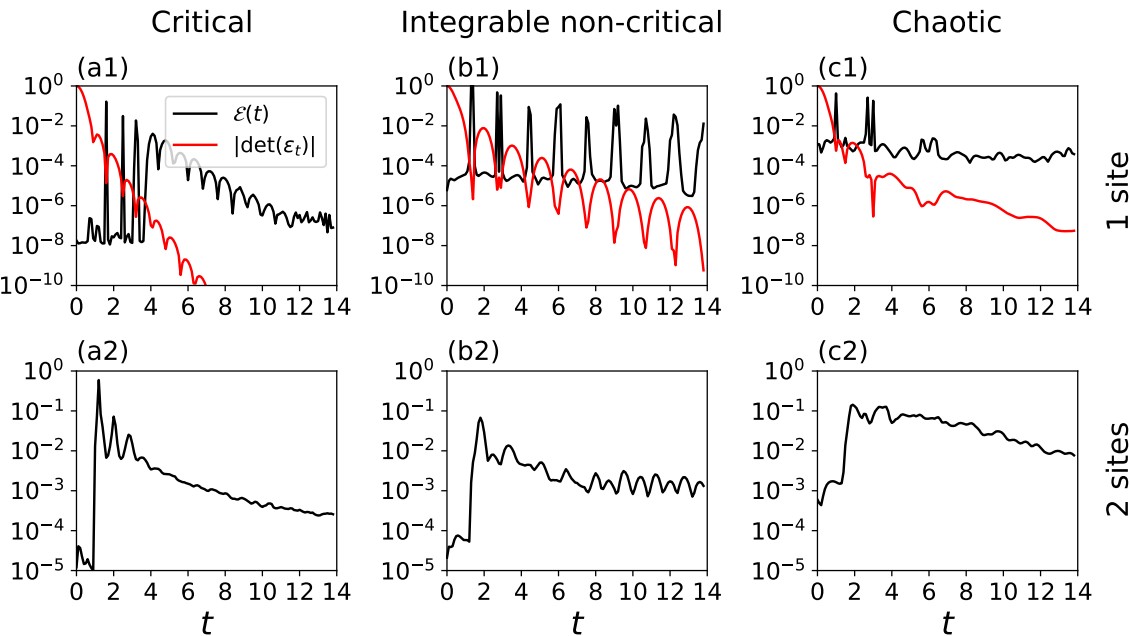

Figure 3: Error function $\mathcal{E}(t)$, quantifying the non-divisibility at different times $t$ for single-site $l = 1$ (first row) and two-site $l = 2$ (second row) subsystems, across the three case studies: (a1,a2) critical ($g^x = 1.0$, $g^z = 0.0$), (b1,b2) integrable non-critical ($g^x = 1.5$, $g^z = 0.0$) and (c1,c2) chaotic ($g^x = -1.05$, $g^z = 0.5$) quantum Ising chains. In red, we plot the absolute value of the determinant of $\epsilon_{t,0}$.

can be reconstructed recursively as

$$\begin{cases} \epsilon^{\text{OPT}}_{t+\text{d}t,0} = (1 + \Lambda^{\text{OPT}}_t \text{d}t)\,\epsilon^{\text{OPT}}_{t,0} \\ \epsilon^{\text{OPT}}_{0,0} = 1 \end{cases} \qquad . \tag{12}$$

In Fig. 3, we plot $\mathcal{E}(t)$ as a function of time for $l = 1$ and $l = 2$ subsystem sites, comparing the critical, integrable non-critical and chaotic regimes. The error function behaviour suggests that the time-local approximation is more appropriate at the critical parameters, with the dynamical map being nearly divisible. As expected, the determinant and the error function exhibit complementary behavior: the error function displays sharp peaks at the times when the dynamical map loses divisibility, followed by a rapid decay back to zero. It is worth emphasizing that for single-site maps at criticality, $|e_3(t)|$ and $|e_4(t)|$ exponentially decay to zero after a short time window [Fig. 2(a)]; nonetheless the error function stays relatively small and consistently decreases [Fig. 3(a1)]. This behavior suggests that, in the critical regime and at long times, the optimal time-local propagator could be found by approximating the dynamical map with a stationary kernel and computing its pseudo-inverse, as proposed in [39]. For the chaotic case, we observe an analogous behavior of the eigenvalues [Fig. 2(c)], which suggests that a pseudo-invertibility will also become possible in this case, only at longer times.

While $\epsilon^{\text{OPT}}_{t,0}$ is expected to be a completely positive trace-preserving (CPTP) map, the loss function in Eq. (10) does not explicitly impose any constraint to guarantee this, potentially leading to instabilities at late times. Even though we typically observe well behaved dynamics, we discuss in Appendix B also an alternative, possibly more stable choice, for the loss function.

The problem of reconstructing the subsystem dynamics in a quantum many-body system has been recently addressed in Refs. [28–30]. In these works, different to our approach, the aim was to learn an effective subsystem dynamics from the values of local observables of support $l = 1, 2$, obtained from some quantum hardware (e.g. quantum computer). In contrast,

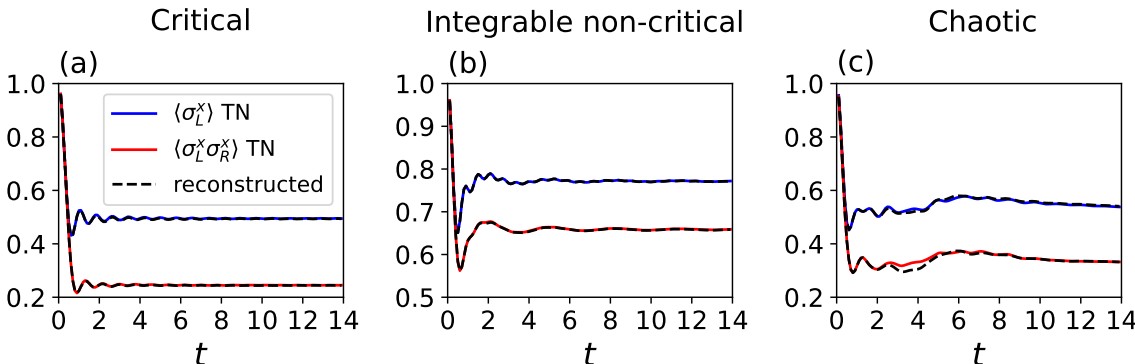

Figure 4: We compare the expectation values $\langle \sigma_L^x \rangle$ and $\langle \sigma_L^x \sigma_R^x \rangle$ obtained from TN simulations (red and blue curves) with those reconstructed by the optimal time-local representations $\Lambda_t^{\mathrm{OPT}}$ (black dashed lines). We present results for two-site subsystems ($l = 2$) across the three case studies: (a) critical ($g^x = 1.0$, $g^z = 0.0$), (b) integrable non-critical ($g^x = 1.5$, $g^z = 0.0$) and (c) chaotic ($g^x = -1.05$, $g^z = 0.5$) quantum Ising chains. The subsystem is prepared into $\hat{\rho}_S(0) = \left| X_L^+ X_R^+ \right\rangle \left\langle X_L^+ X_R^+ \right|$ at time $t = 0$.

our input is the quasi-exact dynamical map $\epsilon_{t,0}$ obtained from some numerical methods that can perform (thermodynamic) calculations up to intermediate times $t$. Another difference is that Refs. [28–30] focus on Lindblad, completely positive parametrization of the subsystem dynamics, while we allow for non-Markovian features that we aim to analyse a posteriori.

## 5 Time-local representations

Even though the error function does not vanish, a relevant question is how well the approximated time-local maps capture the physical observables of the subsystem. To evaluate this, we use the optimized dynamical maps (12) to reconstruct the evolution of local observables, which are then compared to the results from TN simulations. In Fig. 4, we show the expectation values $\langle \hat{\sigma}_L^x \rangle(t)$ and $\langle \hat{\sigma}_L^x \hat{\sigma}_R^x \rangle(t)$ as a function of time $t$, obtained from the subsystem's dynamical maps $\epsilon_{t,0}$ (red lines) and the optimized divisible forms $\epsilon_{t,0}^{\mathrm{OPT}}$ (black dashed lines). As illustrated in this figure, the learned time-local forms effectively capture the subsystem dynamics, particularly in the integrable case. In the chaotic case, we observe larger deviations from ideal behavior, especially in the intermediate times, indirectly suggesting that the dynamics deviate further from divisibility, as also reflected by the error function in Fig. 3. Nevertheless, the late-time dynamics is accurately reconstructed in all three regimes considered: critical, integrable non-critical and chaotic.

Our optimization yields concrete values for the parameters $h_i(t)$ and $c_{ij}(t)$, enabling a direct analysis of the effective Hamiltonian and the dissipative amplitudes $\gamma_i(t)$—the eigenvalues of the coefficient matrix $c(t)$—in the time-local approximation. Importantly, the loss function (10) can have multiple local minima and in practice we find several equivalent time-local representations that yield practically indistinguishable dynamics of local observables.

We present the results of the optimization in Fig. 5. What is common to all the studied regimes (critical, integrable non-critical and chaotic) is that at time $t = 0$, the effective Hamiltonian $\hat{H}_{\mathrm{eff}}(0)$ corresponds to the subsystem Hamiltonian (2), while later it starts to deviate away from it. The noninvertibility of the matrix $\epsilon_{t,0}$ at certain times is most clearly exposed when considering subsystem size $l = 1$, where it is manifested in divergent peaks in $h_i(t)$ and $\gamma_i(t)$. In all cases considered, some of the eigenvalues $\gamma_i(t)$ in the dissipator are negative, which indicates a breakdown of CP-divisibility and signals the non-Markovian nature of the

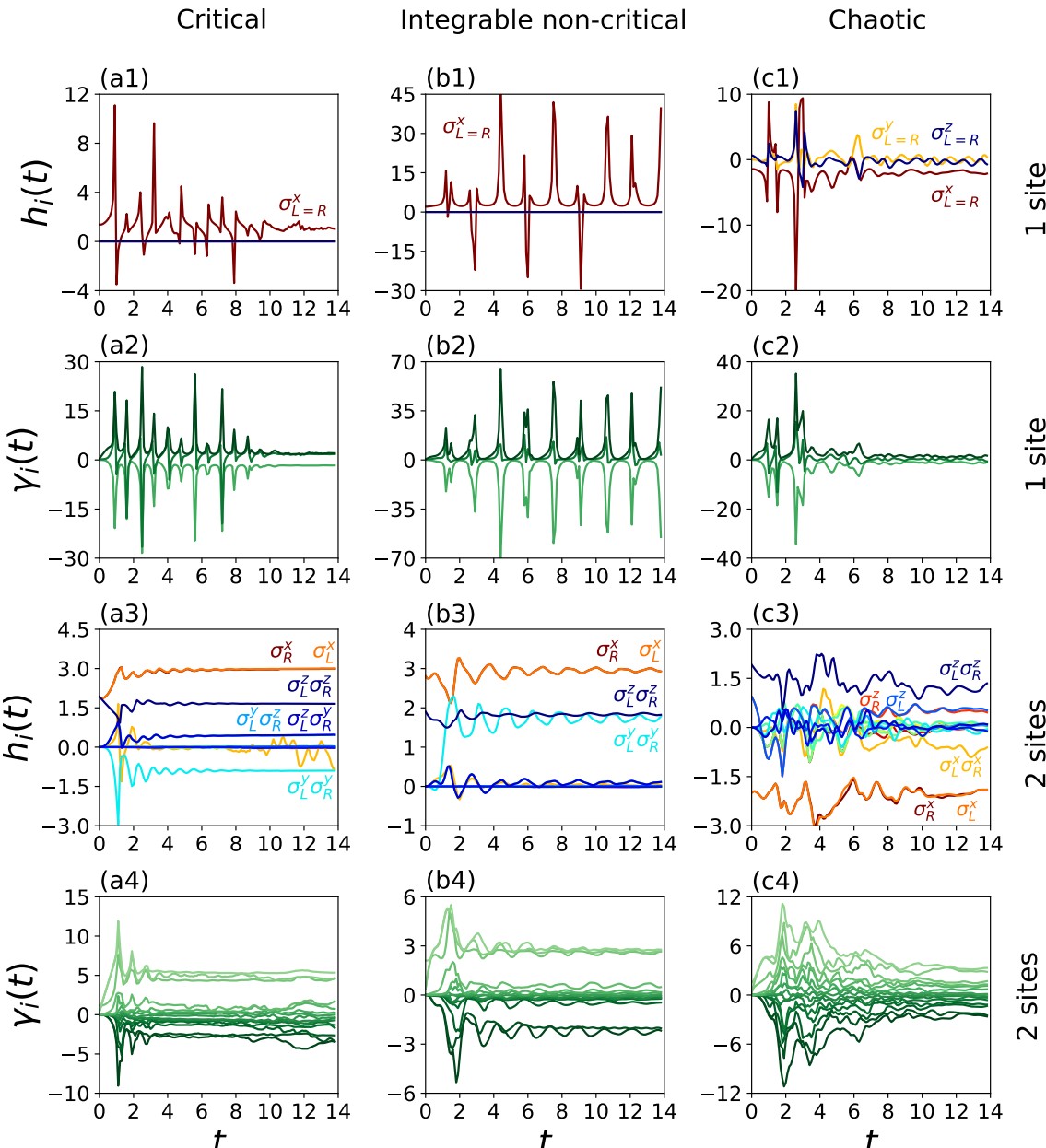

Figure 5: Time evolution of learned Hamiltonian parameters $h_i(t)$ and eigenvalues $\gamma_i(t)$ for subsystems of different support $l = 1$ (first two rows) and $l = 2$ (last two rows) and for different Ising parameters. Left column: critical parameters ($g^x = 1.0$, $g^z = 0.0$). Central column: integrable non-critical parameters ($g^x = 1.5$, $g^z = 0.0$). Right column: chaotic parameters ($g^x = -1.05$, $g^z = 0.5$). In the plots for the coefficients $h_i(t)$ of the effective Hamiltonian, we explicitly highlight the most significant terms, along with their corresponding elements of the Liouville basis, given by tensor products of normalized strings $2^{-1/2}(\hat{\mathbb{1}}_k, \hat{\sigma}_k^x, \hat{\sigma}_k^y, \hat{\sigma}_k^z)$.

subsystem dynamics, according to the Rivas-Huelga-Plenio (RHP) criterion [31–33,36,38,44–52]. While negative eigenvalues $\gamma_i(t)$ may seem unconventional and prevent a straightforward interpretation of dissipative channels as sources of quantum jumps, they remain physically valid. As Fig. 4 shows, they predict the correct time evolution and proper relaxation at long times.

## 5.1 Structure of the time-local approximation

Below, we discuss the results for each regime separately, as illustrated in Fig. 5.

### 5.1.1 Critical case

In the first column of Fig. 5, we present the optimization results for the integrable critical quantum Ising chain ($g^x = 1.0$, $g^z = 0.0$). We plot the Hamiltonian coefficients $h_i(t)$ and the eigenvalues $\gamma_i(t)$ of the Hermitian matrix $\mathbf{c}(t)$ as a function of time, for $l = 1$ and $l = 2$ subsystem sites.

For a single site [Fig. 5(a1,a2)], we find that the effective Hamiltonian $\hat{H}_{\text{eff}}(t)$ depends solely on $\hat{\sigma}^x_{L=R}$. At time $t = 0$, $\hat{H}_{\text{eff}}(0) = \hat{H}_S = \hat{\sigma}^x_{L=R}$ corresponds to the subsystem Hamiltonian as one could have expected. At intermediate times, we observe divergences in the computed Hamiltonian [Fig. 5(a1)] and dissipator [Fig. 5(a2)], reflecting the lack of divisibility of the reduced dynamical map [Fig. 2, Fig. 3]. However, at large times, they both exhibit a quasi-stationary behavior. Moreover, the eigenvalue $\gamma_i(t)$ corresponding to the Lindblad operator $\hat{\sigma}^y_{L=R}$ remains consistently negative, implying a violation of CP-divisibility.

For two-site subsystems [Fig. 5(a3,a4)], the effective Hamiltonian $\hat{H}_{\text{eff}}(t)$ exhibits a much richer structure [Fig. 5(a3)] including terms such as $\hat{\sigma}^y_L \hat{\sigma}^z_R$, $\hat{\sigma}^z_L \hat{\sigma}^y_R$ and $\hat{\sigma}^y_L \hat{\sigma}^y_R$. These non-trivial contributions are absent from the Ising Hamiltonian (1) and do not commute with $\hat{H}_S$, meaning they cannot be interpreted as a Lamb shift, which typically arises in Markovian Lindblad descriptions [16]. Also the Lindblad dissipator exhibits significantly richer structure [Fig. 5(a4)] compared to the single-site case [Fig. 5(a2)]. Specifically, we find 15 distinct eigenvalues, each contributing in a non-trivial way to the overall dynamics of the subsystem, reflecting the interactions and quantum correlations between the subsystem and its environment. Remarkably, $h_i(t)$ and $\gamma_i(t)$ become approximately stationary rather early, compared to the other regimes that we describe below.

### 5.1.2 Integrable non-critical case

The second column of Fig. 5 presents the optimization results for an integrable non-critical quantum Ising chain ($g^x = 1.5$, $g^z = 0.0$). As for the critical case, for single-site subsystems [Fig. 5(b1,b2)], the effective Hamiltonian $\hat{H}_{\text{eff}}(t)$ depends solely on $\hat{\sigma}^x_{L=R}$ [Fig. 5(b1)]. The Lindblad dissipator exhibits a more irregular pattern and discontinuities, even at long times, with one eigenvalue staying consistently negative [Fig. 5(b2)].

For two-site subsystems [Fig. 5(b3,b4)], the effective Hamiltonian $\hat{H}_{\text{eff}}(t)$ again exhibits a much richer structure, including terms that are absent from the Ising Hamiltonian (1) and which do not commute with $\hat{H}_S$ [Fig. 5(b3)]. The Lindblad dissipator continues to display significant complexity [Fig. 5(b4)], however, no discontinuities are observed. Unlike in other regimes, we observe oscillatory behaviour in $h_i(t)$ and $\gamma_i(t)$, which persist up to large simulation times.

### 5.1.3 Chaotic case

In the third column of Fig. 5, we show the results for a chaotic case ($g^x = -1.05$, $g^z = 0.5$). Contrary to the integrable cases, for single-site subsystems [Fig. 5(c1,c2)], the effective Hamiltonian $\hat{H}_{\text{eff}}(t)$ does not only contain $\hat{\sigma}^x_{L=R}$ terms, but is a linear combination of the three basis elements $\hat{\sigma}^x_{L=R}$, $\hat{\sigma}^y_{L=R}$, $\hat{\sigma}^z_{L=R}$ [Fig. 5(c1)]. The Lindblad coefficients $\gamma_i(t)$ exhibit a more irregular time-dependence but, again, one eigenvalue remains negative at all times [Fig. 5(c2)].

For two-site subsystems [Fig. 5(c3,c4)], the effective Hamiltonian $\hat{H}_{\text{eff}}(t)$ again includes terms that are absent from the Ising Hamiltonian (1) and do not commute with $\hat{H}_S$ [Fig. 5(c3)]. In the Lindblad dissipator no discontinuities are observed and, differently from the integrable

cases, no dominant dissipative channels can be clearly identified in the spectrum of the coefficient matrix $c(t)$ [Fig. 5(c4)]. While the transient dynamics is longer, eigenvalues $\gamma_i(t)$ also seem to become eventually stationary.

## 5.2 Forecasting long-time observables

In this section, we discuss whether we can leverage our learning approach into a technique that can be used to obtain subsystem dynamics at times that are not accessible to the full TN calculation. We attempt that by taking the time average of the optimal time-local representation,

$$\overline{\Lambda} := \frac{1}{t_2 - t_1} \int_{t_1}^{t_2} \mathrm{d}s\, \Lambda_s\,, \tag{13}$$

where the integration bounds must be chosen such that the Lindblad operators have relaxed and the Liouvillian is effectively stationary in that interval. In this work, we fix $t_2 = 14$ to the longest time calculated via TN simulations, while $t_1$ is chosen appropriately case by case. Applying this stationary Liouvillian to the TN-calculated subsystem state $\rho_S(t')$ at a time $t' \geq t_1$, we obtain the forecast $\hat{\rho}_S^{\mathrm{f}}(t)$,

$$\hat{\rho}_S^{\mathrm{f}}(t) = e^{\overline{\Lambda}(t-t')}\hat{\rho}_S(t')\,. \tag{14}$$

In Fig. 6, we present results regarding the forecasting of long-time dynamics of single-site and two-site observables. We evolve the state up to time $t_1$ using TNs (gray region). In the time window $t \in [t_1, t_2]$ we benchmark the appropriateness of the averaged $\overline{\Lambda}$ by comparing the quasi-exact expectation values to the ones obtained by the time evolution $\hat{\rho}_S^{\mathrm{f}}(t) = e^{\overline{\Lambda}(t-t_1)}\hat{\rho}_S(t_1)$ with averaged Liouvillian (blue region). Finally, we extend the propagation from $\hat{\rho}_S(t_1)$ beyond the maximal time $t_2$ reached with TN calculations and forecast the behaviour of subsystem observables at $t > t_2$ (green region). For the critical and integrable non-critical quantum Ising chains, we fix $t_1 = 3$, while for the chaotic case, we fix $t_1 = 8$, as the relaxation of time-dependent Lindbladians emerges at longer times [Fig. 5].

### 5.2.1 Critical case

Fig. 6(a2-a4) demonstrates that the averaged $\overline{\Lambda}$ can effectively reproduce the local dynamics at large times for two-site subsystems ($l = 2$) at the critical point ($g^x = 1.0$, $g^z = 0.0$): benchmarking against the quasi-exact results in the blue region is quite reasonable, therefore we can believe that the forecasted expectation values in the green region give the right trend as well. This could have been anticipated from almost constant learned $h_i$ and $\gamma_i$ [Fig. 5(a3, a4)]. Interestingly, divergences in the single-site learned map [Fig. 5(a1, a2)] are apparently a crucial feature that drives the oscillatory dynamics in single-site observables. If using the averaged propagator, oscillations are completely missed [Fig. 6(a1)]. Therefore, in this case, the effect of the environment is more easily reconstructed for the larger subsystem.

### 5.2.2 Integrable non-critical case

Fig. 6(b1-b4) shows that for the integrable non-critical chain ($g^x = 1.5$, $g^z = 0.0$), the Lindblad operators contain too much structured time dependence to be replaced by their long-time average. Indeed, most of the expected features of the observables are completely missed.

### 5.2.3 Chaotic case

Since the transient dynamics is longer in the chaotic case ($g^x = -1.05$, $g^z = 0.5$) averaging is performed on a smaller interval $[t_1, t_2] = [8, 14]$, in turn increasing the averaging error.

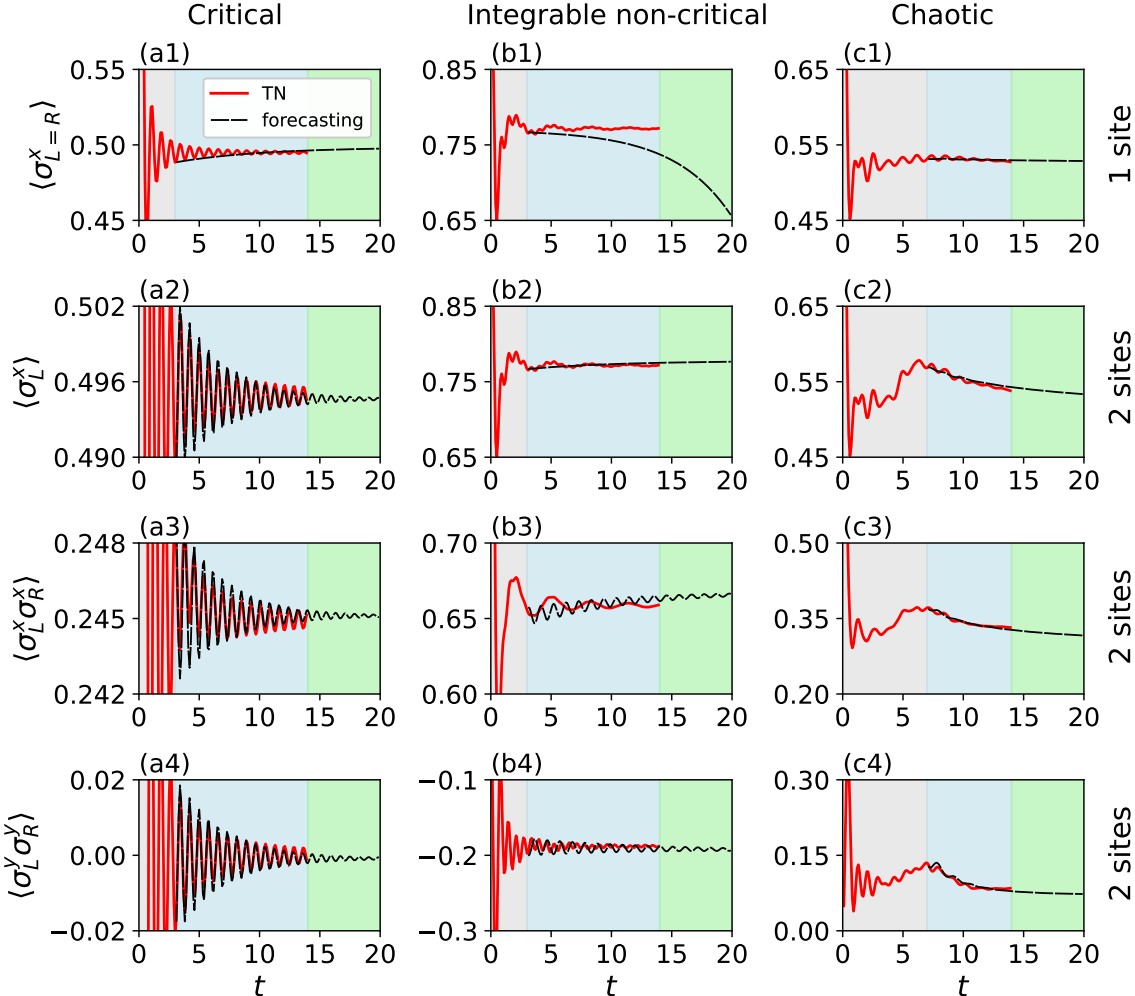

Figure 6: Time evolution of observables from the time-averaged propagator (black dashed lines) vs. the dynamical map $\epsilon_{t,0}$ (red curves). First, we propagate the state up to time $t_1$ using TNs (gray region), $\hat{\rho}_S(t_1) = \epsilon_{t_1,0}[\hat{\rho}_S(0)]$; for $t \in [t_1, t_2]$, we benchmark the solution (14) using the time-averaged Liouvillian $\overline{\Lambda}$ (blue region); for $t > t_2$, we forecast the dynamics of observables (green region). We present results for single-site $l = 1$ (first row) and two-site $l = 2$ (last three rows) subsystems across the three case studies. First column: critical ($g^x = 1.0$, $g^z = 0.0$, $t_1 = 3$) quantum Ising chain. Middle column: integrable non-critical ($g^x = 1.5$, $g^z = 0.0$, $t_1 = 3$) quantum Ising chain. Third column: chaotic ($g^x = -1.05$, $g^z = 0.5$, $t_1 = 8$) quantum Ising chain. The subsystem is prepared into $\hat{\rho}_S(0) = \left| X_{L=R}^+ \right\rangle \left\langle X_{L=R}^+ \right|$ for $l = 1$ and $\hat{\rho}_S(0) = \left| X_L^+ X_R^+ \right\rangle \left\langle X_L^+ X_R^+ \right|$ for $l = 2$ at time $t = 0$. For each row, the considered observable is indicated on the left.

Nonetheless, the two-site averaged $\overline{\Lambda}$ does seem to capture the non-trivial trend in the relaxation of local observables, while it fails to reproduce the fine oscillations, see blue regions in Fig. 6(c1-c4). We thus expect that the forecasting for longer times in the green region is also capturing the general trend of the observable's evolution. Investing more computational effort in the quasi-exact TN evolution should provide data at long enough time to improve the quality of this forecasting for even longer times, as observed in the critical case.

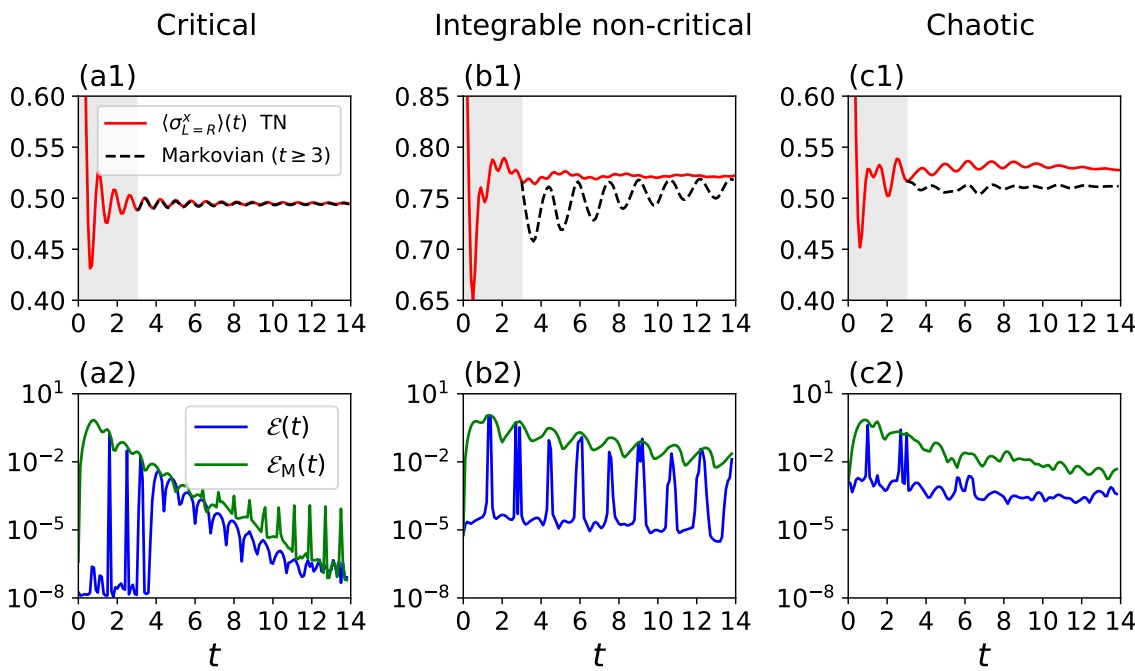

Figure 7: First row: Comparison between the dynamics of $\langle \sigma_L^x \rangle$ (red curves) generated by the dynamical map $\epsilon_{t,0}$ and that reconstructed by the optimal Markovian time-local representation $\mathcal{L}_t^{\text{OPT}}$ (dashed) for the (a1) critical ($g^x = 1.0$, $g^z = 0.0$), (b1) integrable non-critical ($g^x = 1.5$, $g^z = 0.0$) and (c1) chaotic ($g^x = -1.05$, $g^z = 0.5$) parameters on $l = 1$ subsystem size. Second row: Error functions $\mathcal{E}(t)$ and $\mathcal{E}_{\text{M}}(t)$, Eqs. (11,16), at different times for $l = 1$ and (a2) critical, (b2) integrable non-critical and (c2) chaotic parameters.

## 6  Non-Markovianity measures

According to the RHP criterion [31–33, 36, 38, 44–52], a dynamical map is Markovian if and only if it is CP-divisible. For the subsystem dynamics considered in this paper, dynamical maps are singular, i.e., non-divisible, which by definition classifies the dynamics as non-Markovian under the RHP criterion. For this reason, we propose to measure the degree of non-Markovianity based on the distance between the quasi-exact evolution and the closest time-local CP-divisible form. For this reason, we define the following loss function

$$\text{loss}_{\text{M}}(t, \mathcal{L}_t) \equiv \| \dot{\epsilon}_{t,0} - \mathcal{L}_t \, \epsilon_{t,0} \|. \tag{15}$$

Here, $\mathcal{L}_t$ represents any completely positive time-local form, that means $c(t)$ in Eq. (8) has to be positive semi-definite. Therefore, we can parametrize $c(t) = d(t)^\dagger d(t)$ for some complex, time-dependent $(d^2 - 1) \times (d^2 - 1)$ matrix $d(t)$. The optimized CP-divisible form $\mathcal{L}_t^{\text{OPT}}$ is obtained by minimizing the loss function (15) over the space of $(d^2 - 1) \times (d^2 - 1)$ complex matrices $d(t)$ and real vectors $h(t) = (h_1(t), \ldots, h_{d^2-1}(t))$ at any time $t$. The total amount of non-Markovianity up to time $t$ can be measured by using the error function $\mathcal{E}_{\text{M}}(t)$,

$$N(t) \equiv \int_0^t ds \, \mathcal{E}_{\text{M}}(s), \qquad \mathcal{E}_{\text{M}}(t) \equiv \text{loss}_{\text{M}}(t, \mathcal{L}_t^{\text{OPT}}). \tag{16}$$

This method of characterizing non-Markovian effects via the Frobenius distance offers an alternative to recent developments that utilize process tensor methods [53].

After minimizing (15), we can examine the difference between the true time-dependent observables and the result of the closest Markovian form—as previously done for the optimized time-local forms in Fig. 4. In Fig. 7(a1,b1,c1), we compare the expectation values of the observable $\langle \hat{\sigma}_{L=R}^x \rangle$ obtained from the single-site dynamical map $\epsilon_{t,0}$ (red line) and the closest Markovian form (black dashed line). More specifically, to check the accuracy of the Markovian ansatz at large times, we prepared the subsystem into $\hat{\rho}_S(0) = \left| X_{L=R}^+ \right\rangle\left\langle X_{L=R}^+ \right|$ at time $t = 0$, evolve the state up to time $t = 3$ using the dynamical map $\epsilon_{3,0}$ and then propagate the state using the CP form $\mathcal{L}_t^{\text{OPT}}$ for $t \geq 3$. In Fig. 7, we show results for the critical (first column), integrable non-critical (second column) and chaotic (third column) quantum Ising chains. The most significant finding is that, for the critical chain, after a transient short-time regime, the Markovian ansatz accurately captures the oscillatory behavior of the observables, indicating that the dynamics is approaching the Markovian regime. In contrast, this does not hold for the integrable non-critical and chaotic cases, as the Markovian ansatz fails to accurately interpolate the expectation values of the observables. This is further supported by the time evolution of the error functions $\mathcal{E}_M(t)$ and $\mathcal{E}(t)$, Eqs. (11,16), shown in Fig. 7(a2,b2,c2). The blue curve represents $\mathcal{E}(t)$, namely the optimization results in the most general case, in which the matrix $c(t)$ is only required to be Hermitian. In contrast, the green curve corresponds to $\mathcal{E}_M(t)$, namely the results obtained under the Markovian approximation, where the minima are constrained to the space of positive semi-definite Hermitian matrices. The figure illustrates that, at the critical point, after an initial transient regime—during which the Markovian constraint fails to fully capture the subsystem dynamics—the two curves gradually converge, indicating that the system is approaching the Markovian regime. Again, this is not true for the integrable non-critical and chaotic cases, where convergence seems to be much slower.

The lowest non-Markovianity found for the integrable critical model is consistent with the intuitive expectation. In this case, the model is equivalent to a system of free fermions and our initial state corresponds to a non-equilibrium quasiparticle occupation with the largest occupation of the fastest modes. The dynamics is subsequently dominated by the largest velocities of quasiparticles. The folded transverse TN in this case (see also [20]) resembles that of a dual unitary circuit, for which the unentangled boundary vectors are equivalent to a Markovian subsystem dynamics [54].

Next, we test whether our new measure of non-Markovianity for dynamical maps (16) is consistent with two other commonly used measures. In particular, we compare our results with the LPP measure [44] and the BLP measure [45]. The LPP measure evaluates the degree of non-Markovianity from the time-evolution of the volume of dynamically accessible states in the Bloch hypersphere. More specifically, this volume is given by the absolute value of the determinant of the dynamical map $\epsilon_{t,0}$, which always decreases monotonically over time under positive-divisible dynamical maps. Therefore, any non-monotonic behavior indicates non-Markovianity according to the LPP criterion. Following Ref. [44], the LPP measure is defined as

$$N_{\text{LPP}}(t) \equiv \int_0^t \mathrm{d}s\, f_{\text{LPP}}(s)\,, \tag{17}$$

where $f_{\text{LPP}}(s) = \partial_s \left| \det(\epsilon_{s,0}) \right|$ if $\partial_s \left| \det(\epsilon_{s,0}) \right| \geq 0$ and zero otherwise. On the other hand, the BLP measure focuses on distinguishability of quantum states, and is based on the contractivity under CPTP maps of the trace distance, which for two states $\hat{\rho}$ and $\hat{\sigma}$ is defined as $\mathcal{D}(\hat{\rho}, \hat{\sigma}) = \frac{1}{2} \mathrm{tr}\left(\sqrt{\hat{\Delta}^2}\right)$, where $\hat{\Delta} = \hat{\rho} - \hat{\sigma}$. Namely, in the Markovian case, the trace distance between any pair of initial states $\hat{\rho}(0), \hat{\sigma}(0)$, must be a monotonically decreasing function over time. That means

$$\mathcal{D}\left(\epsilon_{\tau_1,0}[\hat{\rho}(0)], \epsilon_{\tau_1,0}[\hat{\sigma}(0)]\right) \geq \mathcal{D}\left(\epsilon_{\tau_2,0}[\hat{\rho}(0)], \epsilon_{\tau_2,0}[\hat{\sigma}(0)]\right)\,, \ \forall \tau_1 < \tau_2, \ \forall \hat{\rho}(0), \hat{\sigma}(0)\,. \tag{18}$$

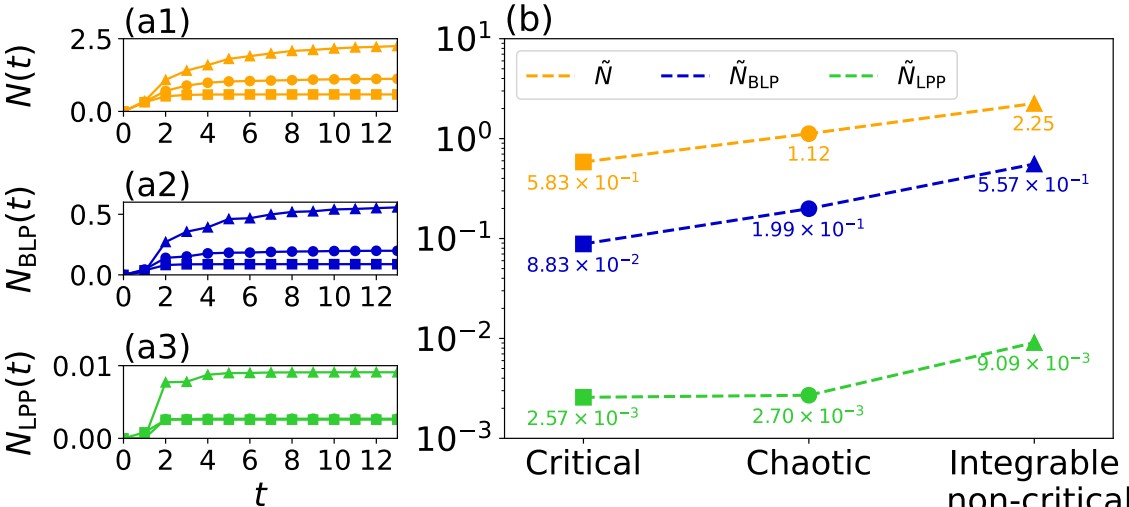

Figure 8: (a1,a2,a3) Time evolution of the different measures of non-Markovianity, defined in Eqs. (16,17,19), for the critical ($g^x = 1.0$, $g^z = 0.0$) (squares), integrable non-critical ($g^x = 1.5$, $g^z = 0.0$) (triangles) and chaotic ($g^x = -1.05$, $g^z = 0.5$) (circles) parameters on $l = 1$ subsystem size. (b) Their values at last (almost stationary) time moment ($t = 13$). Behaviour of our newly defined measure $N(t)$ (orange) is consistent with BLP (blue) and LPP (green) measures.

Any deviation from this behavior signals a revival of coherences, interpreted as information backflow from the environment to the system. This backflow implies the presence of memory effects, as information is temporarily stored in the environment before returning to influence the system at a later time. The BLP measure quantifies non-Markovianity as the maximal backflow of information over all possible pairs of initially orthogonal (i.e. maximally distinguishable) states, namely

$$N_{\text{BLP}}(t) \equiv \max_{\substack{\hat{\rho}(0),\hat{\sigma}(0) \\ \text{tr}[\hat{\rho}(0)\hat{\sigma}(0)]=0}} \int_0^t \mathrm{d}s \, f_{\text{BLP}}(s), \tag{19}$$

where $f_{\text{BLP}}(s) = \partial_s \mathcal{D}(\hat{\rho}(s), \hat{\sigma}(s))$ if $\partial_s \mathcal{D}(\hat{\rho}(s), \hat{\sigma}(s)) \geq 0$ and zero otherwise, with maximization done over any pair of mutually orthogonal initial states, $\text{tr}[\hat{\rho}(0)\hat{\sigma}(0)] = 0$.

Fig. 8 illustrates the time evolution of these measures and their long-time values for the three models studied and the case of a single-site subsystem ($l = 1$). All measures have been computed numerically; in particular, the BLP measures has been computed by a stochastic sampling of $5 \times 10^4$ pairs of orthogonal initial states. Notably, after an initial transient time all measures show consistent results: (i) Non-Markovian effects are largest for the integrable non-critical case and smallest for the critical dynamics. (ii) The largest contribution to the non-Markovianity happens at intermediate times. These results support the newly introduced measure (16), as they show its consistency with other criteria.

The results presented in this section pertain only to single-site subsystems. Although similar conclusions are expected for $l = 2$, the number of ADAM optimization epochs appears insufficient to clearly observe this behavior. We leave a more thorough analysis of this case for future work.

# 7 Conclusions

In this work, we characterized the subsystem dynamics of a coherent Hamiltonian evolution. We considered a partition of an infinite quantum Ising chain, defining a subsystem consisting of an $l$-site Ising chain coupled to two semi-infinite Ising chains that act as the external environment. The subsystem's dynamical maps are computed using TN simulations, specifically, transverse light-cone contractions, which provide quasi-exact results for intermediate times. The spectrum of the dynamical maps shows that they become manifestly non-invertible after short times. This lack of invertibility indicates that the subsystem dynamics are governed by non-trivial memory kernels, and it is not possible to obtain exact time-local forms. Our efforts focus instead on learning the optimal time-local representations that best capture the subsystem dynamics, a task that we achieved through the ADAM optimization algorithm. More specifically, we considered $l = 1$ and $l = 2$ subsystem sites and we explored different regimes (critical integrable, integrable and chaotic), all generically characterized by non-Markovian dynamics, driven by non-stationary environments and strong system-bath couplings.

The results of this optimization present a number of interesting features. First, even though the full dynamical maps are not divisible, the optimal time-local representations are able to reproduce the time-dependent local observables along practically the full evolution, with the only exception of a small short-time window in the non-integrable case. Second, we quantified the deviation of the optimal divisible dynamical maps from the primary ones obtained via the transverse folding strategy, and observed that, in general, the error decreases over time, indicating that the subsystem's evolution increasingly better approximates a time-local form. This decay is fastest in the Ising critical case and slowest in the integrable non-critical case.

Then, we analyzed the structure of the approximate time-local forms extracted from our optimization. Single-site subsystems typically exhibit more diverging profiles in the effective Hamiltonian coefficients and Lindblad jump operators. In contrast, the local representations for two-site systems show smoother behavior, likely because of the increased number of parameters involved in the optimization. Importantly, on the level of observables, dynamics generated by the learned maps agrees well with TN results.

For the critical and chaotic transverse field Ising model, the learned time-local forms after some transient dynamics become almost stationary. We showed that the time-averaged propagator, extracted from the quasi-stationary regime, can be used to extrapolate (the main) relaxation features of local observables to times that exceed what is simulable with TN techniques. This suggests our learning approach could be used to perform longer subsystem time evolution. However, our forecasting of observables fails for the integrable non-critical dynamics, where the learned time-local form retains too much of structured time dependence that cannot be neglected and is difficult to extrapolate to longer times.

Finally, we explored how closely the dynamical map can be approximated by a Markovian form. We found that at the critical point the dynamics converges toward the Markovian limit. Our optimization in this case reproduces the local observables, whereas this is not the case for the non-critical integrable and the chaotic cases, which remain far from a Markovian description at all times. We introduced a new measure of non-Markovianity based on the distance between the quasi-exact dynamical map and the closest CP-divisible form. This measure has been compared with the LPP and BLP measures, showcasing consistent results. However, our results call for further investigation into the generality of these observations —specifically, whether the same conclusions extend to larger subsystems, as expected.

Our work opens a number of directions for possible future studies. Further investigations could provide a deeper understanding of how the features of our learned subsystem description depend on the class of model and the state of environment considered. In particular, one could prepare the environment in different initial states and analyze how their energy density

(effective temperature) affects the observed subsystem behavior. The procedure can also be immediately extended to cases where the environment is in a mixed state, including one of true thermal equilibrium. What also remains to be explored is whether critical dynamics are, in general, more Markovian than non-critical ones, or if this is merely a feature of the specific physical model under consideration.

## Acknowledgements

We acknowledge discussions with Igor Lesanovsky, Federico Carollo, Marcel Cech, Marko Žnidarič, Luca Tagliacozzo, Sarang Gopalakrishnan and Alessandra Colla. MC and ZL were supported by the QuantERA II JTC 2021 grants QuSiED and T-NiSQ by MVZI, the P1-0044 program of the Slovenian Research Agency and ERC StG 2022 project DrumS, Grant Agreement 101077265. MCB was supported by the Deutsche Forschungsgemeinschaft (DFG, German Research Foundation) under Germany's Excellence Strategy – EXC-2111 – 390814868; Research Unit FOR 5522 (grant nr. 499180199), and the EU-QUANTERA project TNiSQ (BA 6059/1-1). ZL and MCB acknowledge support by the Erwin Schrödinger International Institute for Mathematics and Physics (ESI), during the thematic program "Entanglement in Many-body Quantum Matter".

We also gratefully acknowledge the High Performance Computing Research Infrastructure Eastern Region (HCP RIVR) for funding this research by providing computing resources of the HPC system Vega at the Institute of Information sciences.

## A   Optimization analysis

As expected, optimization improves with the number of ADAM steps (epochs). Fig. 9(a1,b1,c1) illustrates what error function $\mathcal{E}(t)$, Eq. (11), we get at different times, depending on the number of ADAM optimization steps for $l = 1$, across three representative cases: the critical, integrable non-critical, and chaotic regimes. For integrable non-critical and chaotic parameters, the error function is rather converged with the number of ADAM optimization steps performed, i.e., the error function does not further decrease for larger number of epochs. For the critical parameters, increasing the number of epochs from $1.5 \times 10^4$ to $3 \times 10^4$ decreases the error function even further. However, these improvements are obtained in the regime where the absolute value of the error function is already small, therefore they would be hardly visible in any local observable. Fig. 9(a2,b2,c2) shows the same study for $l = 2$, with the error function quite converged for the number of ADAM optimization steps considered.

Depending on the subsystem size $l$, the number of real parameters to be estimated is $4^l(4^l - 1)$. This exponential dependence implies that larger subsystems generally require more epochs for successful optimization. Corresponding increase in the computational time practically limits the scalability of our approach to larger subsystem sizes. Throughout this work we aim for the best trade-off between the accuracy of the Lindblad dissipation channels and the overall optimization time.

## B   Alternative loss function

As emphasized in Sec. 4, different loss functions can be considered. Although the loss function (10) is formally the most suitable for estimating the propagator from time $t$ to time $t + dt$, i.e. $\epsilon_{t+dt,t} = 1 + \Lambda_t dt$, its minimization does not impose any constraints on the entire prop-

agation from time 0 to $t$, namely the distance between the estimated dynamical map $\epsilon_{t,0}^{\text{OPT}}$, Eq. (12), and $\epsilon_{t,0}$. For example, the dynamical map $\epsilon_{t,0}$ could be far from divisible at a certain time $\bar{t}$ and, since the optimization process does not account for this information at times $t > \bar{t}$, a small value of $\mathcal{E}(t)$ at times $t > \bar{t}$ would not necessarily imply that $\|\epsilon_{t,0} - \epsilon_{t,0}^{\text{OPT}}\|$ is also small. An alternative definition of loss function that takes into account the deviation of the computed approximation from the actual map at each time is for example

$$\text{loss}_2(t, \Lambda_t) \equiv \|\epsilon_{t+dt,0} - (1 + \Lambda_t dt)\epsilon_{t,0}^{\text{OPT}}\|. \tag{B.1}$$

The best time-local operator $\Lambda_t^{\text{OPT}}$ is now obtained by minimizing the loss function (B.1) over the Hermitian coefficient matrices $\mathbf{c}(t) = \mathbf{c}(t)^\dagger$ and real vectors $\mathbf{h}(t)$ at any time $t$. We define the error function

$$\mathcal{E}_2(t) \equiv \text{loss}_2(t, \Lambda_t^{\text{OPT}}), \tag{B.2}$$

which is the minimum of the loss function (B.1) at fixed time $t$. In Fig. 10, we plot the error function $\mathcal{E}_2(t)$ as a function of time for $l = 1, 2$ and the three case studies: the critical, integrable non-critical and chaotic quantum Ising chains. While the error function $\mathcal{E}(t)$, Eq. (11), has the trend of getting smaller with time, signaling better divisibility of dynamics at longer times, the error function $\mathcal{E}_2(t)$ is becoming stationary at long times. This means that after some time window we can approximately reconstruct the local propagator, without further decreasing the distance to the real map.

The recursive definition (B.1) has the advantage of minimizing the distance between $\epsilon_{t,0}^{\text{OPT}}$ and $\epsilon_{t,0}$ at any time $t$. However, the optimized $\Lambda_t^{\text{OPT}}$ from Eq. (B.1) is inevitably influenced by previous optimization steps and cannot be strictly interpreted as the propagator from time

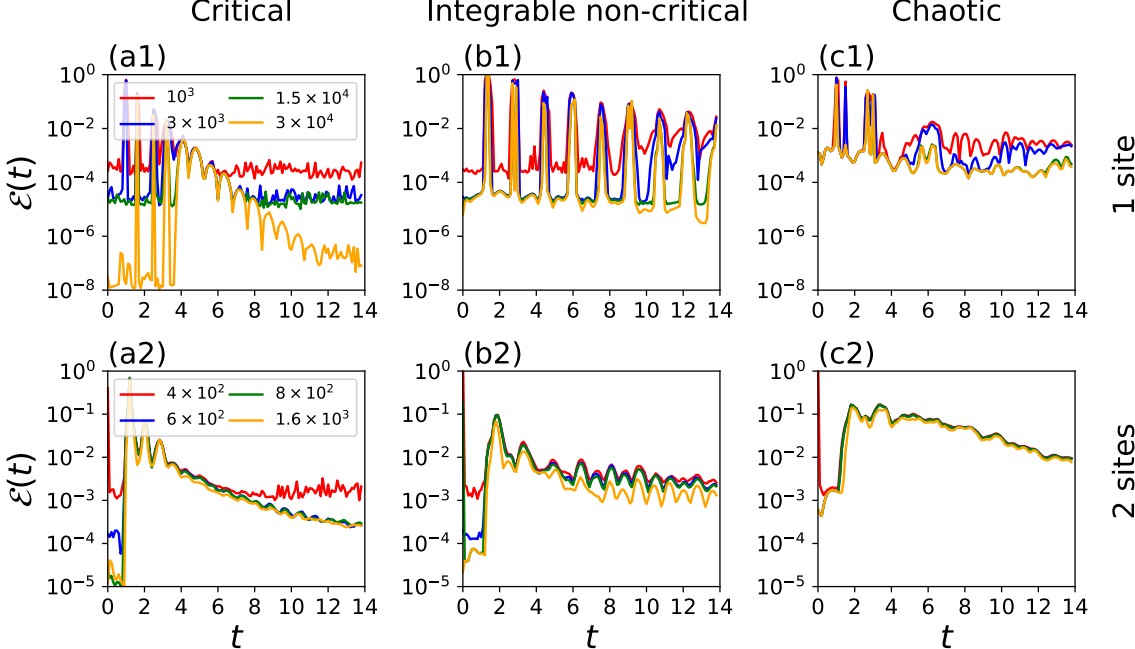

Figure 9: Error $\mathcal{E}(t)$ as a function of time in a semi-log scale, for different number of ADAM optimization steps (epochs) $N_{\text{epochs}} = 10^3, 3 \times 10^3, 1.5 \times 10^4, 3 \times 10^4$. Here, we present the results for single-site ($l = 1$) and two-site ($l = 2$) subsystems, for the (a1,a2) critical ($g^x = 1.0$, $g^z = 0.0$), (b1,b2) integrable non-critical ($g^x = 1.5$, $g^z = 0.0$) and (c1,c2) chaotic ($g^x = -1.05$, $g^z = 0.5$) quantum Ising chains.

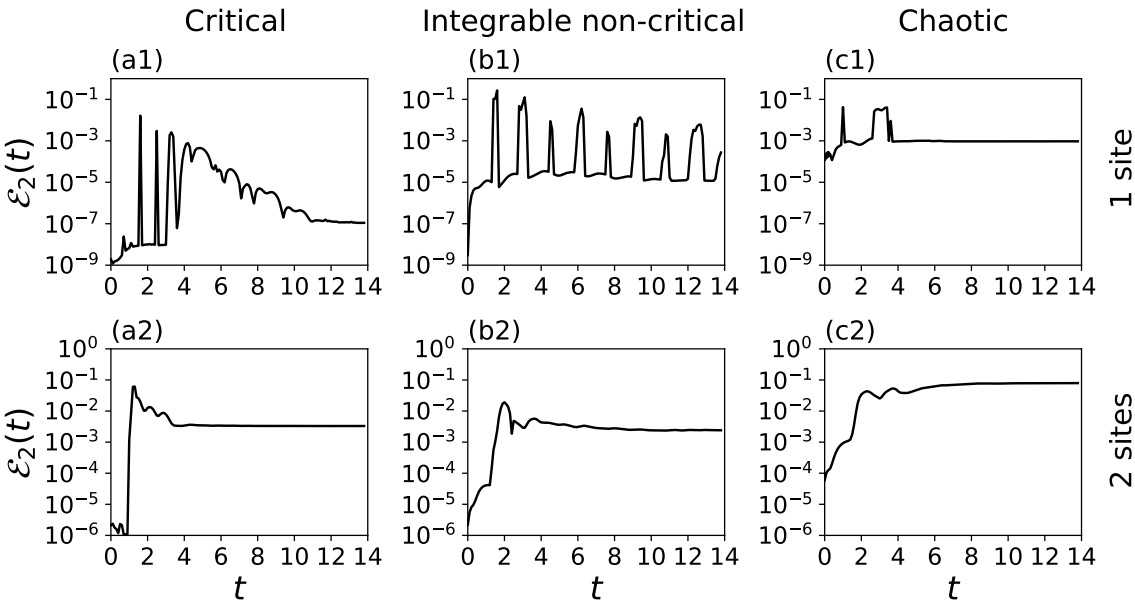

Figure 10: Error $\mathcal{E}_2(t)$ as a function of time $t$ for single-site ($l = 1$) (first row) and two-site $l = 2$ (second row) subsystems, across the three case studies: (a1,a2) critical ($g^x = 1.0$, $g^z = 0.0$), (b1,b2) integrable non-critical ($g^x = 1.5$, $g^z = 0.0$) and (c1,c2) chaotic ($g^x = -1.05$, $g^z = 0.5$) quantum Ising chains.

$t$ to time $t + dt$ as it contains inherent memory effects. Therefore it cannot be used to characterize the emergent Markovianity at longer times, as was done for the other choice of the loss function in the main text, Sec. 6. Since both loss functions (10, B.1) lead to comparably good reconstructions of observables, as shown in Fig. 11, and given that this work primarily focuses on time-local propagators, we used the definition (10) in the main text.

In general, establishing a comprehensive set of minimal criteria for the coefficient matrix $c_{ij}(t)$ to ensure that the Liouvillian (8) preserves the positivity of subsystem states at all times—while also guaranteeing that the full dynamical map $\epsilon_{t,0} = \mathcal{T} \exp\left\{\int_0^t ds \Lambda_s\right\}$, with $\mathcal{T}$ denoting the time-ordering operator, is completely positive—remains a theoretical challenge. When minimizing the loss function (10), no additional constraints are imposed on the matrix $c_{ij}(t)$ beyond enforcing Hermiticity and trace preservation. As a result, the optimal representation may slightly deviate from a truly positive pseudo-Lindblad form. These small deviations can accumulate over time, potentially causing the density operator to drift away from being positive semi-definite at long times. Since the alternative loss function (B.1) looks for the closest divisible form to the completely positive $\epsilon_{t,0}$, it seems to compensate for non-physical contributions that may arise from previous optimization steps. However, for both choices of loss function no practical difficulties were encountered in the time interval considered. Of course, for the Markovian propagator the $c_{ij}(t)$ matrix is positive semi-definite by construction and CP-divisibility is automatically guaranteed, as illustrated in Sec. 6.

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
