# Peer review of "Learning the non-Markovian features of subsystem dynamics"

_SciPost Physics_

## Round 1 · Referee Report · Anonymous (Referee 1) · 2025-9-17

Strengths

  1. Provides a detailed and accurate data shedding new light on the efficient long-time simulation of local observables in quantum systems.
  2. Raises new and interesting questions (for example - the remarkable accuracy of the Markovian approximation to the influence functional in certain cases)
  3. Is clearly written and makes transparent connection to the preceding literature in the field.

Weaknesses

NA

Report

The is a topical and substantial piece of work containing mainly numerical analysis of the accuracy of the Markovian approximation to the influence function in 1d systems. It addresses a set of outstanding questions in a clear and transparent way and raises some interesting questions for future analysis.

The results themselves are remarkable for the accuracy with which this Markovian approximation captures the dynamics of local observables. Perhaps this was not surprising to the authors - it is to me.

I have no criticism of the work per se, but it does raise a number of questions for me that the authors may consider discussing. That the work raises such questions is a strength rather than a weakness

Requested changes

the authors might consider adding a brief discussion of the following points: 1. Once the influence function settles, a constant approximation to it for later times does a good job of approximating local observables. I would have liked a little more discussion about the nature of this steady-state influence function: does it characterise the influence of a thermal system at the energy density considered? 2. On a related point, the authors mention that they actually find multiple fixed points the choice between which does not affect local observables. A little more discussion of the nature of these different influence functions would be instructive - should they be thought of as different parametrizations of the influence of the same steady state or is something else going on? 3. The information lattice work of [12], suggests (if I understand correctly) that retaining information on a slightly longer scale than that of interest might be sufficient. In this context it might be useful to show the single site observables in the l=2 case especially in the non-integrable case in the region where otherwise the simulation is slightly off

Recommendation

Publish (surpasses expectations and criteria for this Journal; among top 10%)

  • validity: top
  • significance: high
  • originality: high
  • clarity: top
  • formatting: perfect
  • grammar: perfect

Author:  Michele Coppola  on 2025-11-04  [id 5984]

(in reply to Report 1 on 2025-09-17)

Dear Editor,

Please find attached our reply to the referee's comments.

Yours sincerely,
Michele Coppola

Attachment:

referee1.pdf

---

## Round 1 · Referee Report · Anonymous (Referee 2) · 2025-10-6

Strengths

The paper is clearly written and introduces a highly relevant and original idea to the field.

Weaknesses

-Certain final results, particularly the prediction of long-time observables, do not achieve very high accuracy

Report

I believe the paper clearly deserves publication in SciPost, because of its fresh, novel ideas as well as its clear presentation. I only provide a few points below for the authors to consider and possibly address in the manuscript.

-You write "a dynamical map is Markovian if and only if it is CP-divisible", while in https://arxiv.org/pdf/1901.05223 the authors claim "We show that completely positive (CP) divisible quantum processes can still involve non-Markovian temporal correlations." Can you comment on that? Moreover, while it may be obvious, why does the non-invertibility of the map imply that it is not divisible? And why negative gamma's imply no CP divisibility?

-"If the dynamical map is invertible, memory-kernel master equations can be rewritten in the time-local form..." To get this form, should the dynamical map be invertible at any time? If the map is not invertible at time t0, can one still get a local form for time t<t0 and t>t0?

-As far as I understood, the dynamical map is obtained explicitly, i.e. as a matrix, after the transverse TN contraction. If it is the case, why you constraint yourself to use the Frobenius norm, while I think other norms are better indicated for operators / channels? For instance norm-1, or the diamond norm. By the way, while obvious for TN experts, the acronym MPO at page 6 is never defined.

-If I understand correctly, a separate optimization must be performed for each time t of interest. Is that right?

-How the initial values of the parameters c(t) and h(t) are chosen? Is the learning procedure robust with respect to this initialization? More generally, is the learning procedure robust? For instance, does the optimization reliably converge?

-I didn't quite understand whether the optimization process discussed around Eq.~(15) is the same as the one described around Eq.~(10), and if not, how the two differ.

-Do you have an explanation for why non-Markovianity in Fig.8 is higher in the integrable non-critical case than in chaotic one? Naively, one might expect the opposite.

Recommendation

Publish (surpasses expectations and criteria for this Journal; among top 10%)

  • validity: high
  • significance: high
  • originality: high
  • clarity: high
  • formatting: good
  • grammar: excellent

Author:  Michele Coppola  on 2025-11-04  [id 5985]

(in reply to Report 2 on 2025-10-06)

Dear Editor,

Please find attached our reply to the referee's comments.

Yours sincerely,
Michele Coppola

Attachment:

referee2.pdf

---

## Editorial Decision

accepted_in_target_journal